# Anastomosis Complications after Bronchoplasty: Incidence, Risk Factors, and Treatment Options Reported by a Referral Cancer Center

Lara Girelli [1,*], Luca Bertolaccini [1], Monica Casiraghi [1,2], Francesco Petrella [1,2], Domenico Galetta [1,2], Antonio Mazzella [1], Stefano Donghi [3], Giorgio Lo Iacono [1], Andrea Cara [1], Juliana Guarize [3,†] and Lorenzo Spaggiari [1,2,†]

1    Division of Thoracic Surgery, IEO, European Institute of Oncology, IRCCS, 20141 Milan, Italy; luca.bertolaccini@ieo.it (L.B.); domenico.galetta@ieo.it (D.G.); antonio.mazzella@ieo.it (A.M.); giorgio.loiacono@ieo.it (G.L.I.); andrea.cara@unimi.it (A.C.)
2    Department of Oncology and Hematology-Oncology, University of Milan, 20141 Milan, Italy
3    Interventional Pneumology Unit, IEO, European Institute of Oncology, IRCCS, 20141 Milan, Italy; stefano.donghi@ieo.it (S.D.); juliana.guarize@ieo.it (J.G.)
*    Correspondence: lara.girelli@ieo.it; Tel.: +39-0257489665
†    These authors contributed equally to this work.

**Abstract:** Background: Sleeve lobectomy with bronchoplasty is a safe surgical technique for the management of lung cancer and endobronchial localization of extrapulmonary cancers. However, anastomotic complications can occur, and treatment strategies are not standardized. Methods: Data from 280 patients subjected to bronchoplasty were retrospectively analyzed, focusing on surgical techniques, anastomotic complications, and their management. Multivariate analysis was performed, and Kaplan–Meier curves were used to determine survival. Results: Ninety percent of 280 surgeries were for lung cancer. Anastomotic complications occurred in 6.42% of patients: late stenosis in 3.92% and broncho-pleural fistula in 1.78%. The median survival was 65.90 months (95% CI = 41.76–90.97), with no difference ($p = 0.375$) for patients with (51.28 months) or without (71.03 months) anastomotic complications. Mortality at 30 days was higher with anastomotic complications (16.7% vs. 3%, $p = 0.014$). Multivariable analysis confirmed pathological stage (N+) as a risk factor for anastomotic complications ($p = 0.016$). Our mortality (3.93%) and morbidity rate (41.78%) corresponded to recent series results. Conclusions: In our experience, surgery is preferred to avoid life-threatening complications in bronchopleural fistulas. Bronchoscopic balloon dilatation is preferred for benign strictures. The nodal stage is related to complications ($p = 0.0014$), reflecting the aggressiveness of surgery, which requires extended radical lymphadenectomy.

**Keywords:** bronchoplasty; sleeve resections; bronchoscopy; anastomotic complications; lung cancer

## 1. Introduction

Bronchoplasty is widely accepted as a safe surgical technique that involves parenchymal-sparing resection in patients with lung cancer [1,2].

It was first described in the 1950s as an option for patients with reduced respiratory capacity who were typically excluded from surgery [3,4].

The mortality, morbidity, and 5-year survival rates have gradually improved with the modernization of surgical techniques.

Sleeve lobectomy has been increasingly used for patients with centrally located lung carcinomas that demonstrate bronchial infiltration by advanced mediastinal nodal involvement.

According to the safety and effectiveness of the procedure, indications of its feasibility have been modified in recent years. This technique was first reserved for patients with malignant lung cancer but is now the first choice for low-grade malignancies such as carcinoids and endobronchial neoplasms of extrapulmonary origin [5–9].

Sleeve resection can be offered to individuals with lung cancer in the early stages of I to IIb and either N0 or N1 (hilar) nodal disease. Selected individuals with N2 disease can benefit from this surgery after successful downstaging with induction treatment.

Thanks to technological advancement, complex lung resections such as bronchoplasty can now be carried out with minimally invasive approaches due to improvements in video-assisted thoracic surgery (VATS) and robotic surgery platforms [10–12].

Since combined therapies found a role in everyday practice, numerous studies were conducted to investigate the complications related to bronchoplasty performed after induction chemotherapy or radiotherapy, confirming acceptable mortality and morbidity rates [13–23].

Although this procedure is safe if performed by expert hands, anastomotic complications can occur.

Complications include anastomotic dehiscence, stenosis, and broncho-vascular and broncho-pleural fistulas [24].

Treatments for complications of bronchial anastomosis are not standardized and depend on the experience of different centers [24–26].

Despite the possibility of severe and life-threatening conditions due to the failure of anastomosis, many studies on outcomes and survivals encourage sleeve resections compared to pneumonectomy [27–31].

Endoscopic procedures and re-operations are alternative procedures to treat different types of injuries. The choice depends on the patient's general status, the extension of anastomotic damage, and the specialist's expertise.

An in-depth understanding of the anatomy and physiology of the bronchus is crucial for any comprehensive discussion on bronchoplasty. The bronchus, a primary airway within the respiratory system, exhibits a complex structure and function. Surgeons must grasp basic anatomical features, such as branching patterns, epithelial lining, and supportive cartilage rings. Furthermore, a comprehensive exploration of the physiological processes associated with the bronchus, including ventilation and gas exchange, forms the foundation for appreciating the significance of bronchoplasty. Bronchoplasty, a surgical procedure involving the reconstruction or repair of the bronchus, demands a nuanced comprehension of the intricacies involved. Surgeons must underscore the importance of thoroughly examining CT scans and endoscopic features to elucidate the complications associated with bronchial disorders. Illustrating these features not only enhances the readability of the material but also provides a visual aid for readers to correlate theoretical knowledge with practical applications. CT scans allow for a detailed depiction of structural abnormalities, while endoscopic visuals offer insights into the real-time conditions within the bronchial passages. By integrating these imaging modalities into the discussion, thoracic surgeons can effectively convey the clinical relevance of bronchoplasty, thereby fostering a more engaging and informative reading experience.

Moreover, surgeons should strategically pique readers' interest by highlighting the technical intricacies of bronchoplasty. This involves elucidating the surgical techniques, instruments, and step-by-step procedural aspects. Concomitantly, it is imperative to address the contraindications associated with bronchoplasty. Surgeons should consider the patient selection criteria, considering factors such as comorbidities, overall health status, and the nature of bronchial pathology. A thorough understanding of contraindications is pivotal in ensuring the safety and efficacy of the procedure, and this information should be presented with clarity and precision. Lastly, the feasibility of bronchoplasty should be discussed in a multidisciplinary setting, shedding light on the practical aspects of implementing such a procedure.

This study aims to analyze the results of an extensive series of bronchoplastic procedures for malignant neoplastic disease, focusing on anastomotic complications and the performed treatments. The incidence and risk factors related to anastomosis failure are analyzed and treatment options are presented.

## 2. Materials and Methods

Two hundred and eighty patients underwent sleeve resections for cancer between January 1998 and May 2018 at the European Institute of Oncology (IEO) IRCCS, Milan, Italy.

Patients who underwent bronchoplastic procedures were included, and data were retrospectively collected from hospital archives, operative reports, and medical records. Patients that underwent tracheal resections were excluded from the study.

The preoperative staging included computed tomography (CT) of all patients' chest and abdomen and CT or magnetic resonance of the brain.

The mediastinal invasive staging was performed with cervical mediastinoscopy in the case of radiological evidence of N2 station involvement. In recent years, with the introduction of endobronchial ultrasound (EBUS), the histologic mediastinal staging was extended to all patients. Bronchoscopy was performed for endobronchial evaluation before surgery or intraoperatively if necessary.

The preoperative functional evaluation included pulmonary function testing, arterial blood gas, electrocardiography, and echocardiography for high-cardiological-risk patients.

### 2.1. Surgical Techniques

A double-lumen tube is placed during the induction of anesthesia, and the patient is positioned for a lateral thoracotomy. Muscle-sparing lateral thoracotomy is used to enter the fourth or fifth intercostal space. The lung must be fully mobilized. The pulmonary arterial and venous structures to the interested lobe are isolated, followed by completing the pulmonary fissures.

The subcarinal space is dissected, and the lymph node packet is removed. Carina is well exposed, and the esophagus is mobilized. The distal trachea is gently mobilized with the preservation of vascular support. The airway can be resected with a wedge bronchotomy if cancer involves the origin of the lobar bronchus or a perpendicular cut of the complete circumference when cancer involves the main bronchus. The specimen is sent for the frozen section examination of the resection margins. Once the lobe is removed, release maneuvers are performed to reduce tension on the bronchoplasty.

An intraoperative bronchoscopy can be helpful in assessing reconstructed airway anatomy.

When sleeve lobectomy is performed, the airway reconstruction choices are either interrupted or continuous suture techniques. Authors preferred single- or double-running 3-0 monofilament sutures. Once the anastomosis is completed, it is tested for any air leaks. The transposition of a pedicle flap of pericardial fat, pericardium, or pleura is an excellent option to cover every bronchoplastic technique.

### 2.2. Bronchoscopic Procedures

Bronchoscopy was performed routinely preoperatively to study bronchial anatomy and plan the resection. The procedure was usually repeated intraoperatively to assess the airway's reconstructed continuity. Endoscopic control of the anastomosis was routinely performed before discharge and during the early follow-up, generally at 45 days.

### 2.3. Statistical Analysis

Quantitative variables were expressed as mean $\pm$ standard deviation (SD), whereas nominal variables were expressed binarily as the presence or absence of the event. Recurrence was diagnosed based on imaging or pathological examinations found in a review of the case records. Kruskal–Wallis rank test was used for continuous variables, with Fisher's exact test for categorical variables. Progression-free survival (PFS) and overall survival (OS) were calculated based on the interval between TET resection and detection of recurrence using the Kaplan–Meier method. A backward stepwise Cox regression model was performed to determine the hazard ratio (HR) of factors associated with long-term outcomes. Variables with a $p$-value less than 0.2 were used for multivariate analysis, and significance was defined as a $p$-value less than 0.05. Statistical analyses were performed using R software (version 3.6.1, Action of the Toes; R Foundation for Statistical Computing,

Vienna, Austria) with standard, Rcmdr (version 3.6.1, Action of the Toes; R Foundation for Statistical Computing, Vienna, Austria), and IRR packages (version 3.6.1, Action of the Toes; R Foundation for Statistical Computing, Vienna, Austria).

## 3. Results

Over the 20-year period, 280 patients underwent bronchial sleeve resection and bronchoplasty for cancer in our institute. The mean age was $62.2 \pm 11.6$ years with a range of 18 to 80 years. There were 206 (73.5%) male patients and 74 (26.5%) female patients, with a 2.8 M/F ratio.

Two hundred and thirty (82%) patients were former smokers. Comorbidities included cardiovascular diseases in 98 patients (35%) and pulmonary diseases in 60 patients (21.4%). The preoperative FEV 1 mean was 2.3 L (SD 0.7).

Table 1 shows the population's characteristics. The histologic diagnoses included adenocarcinomas in 90 cases (32.1%), squamous cell carcinoma in 120 patients (42.9%), and carcinoids in 30 patients (10.7%). Small cell carcinoma was found in 11 patients (3.9%). From 1998 to 2008, these patients were operated on under a different preoperative diagnosis. Twenty-nine cases (10.4%) had other histological subtypes, including large cell neuroendocrine carcinomas, sarcomas, metastases, and tumors of combined histology.

**Table 1.** Patient profiles.

| Characteristic | Patients *n* (%) |
|---|---|
| Age mean $\pm$ SD | $62.2 \pm 11.6$ |
| Male/Female ratio | 2.8 |
| Smoking history | 230 (82.1%) |
| Cardiovascular history | 98 (35%) |
| Pulmonary history | 60 (21.4%) |
| Preoperative FEV1 mean $\pm$ SD | $2.3 \pm 0.7$ |
| Preoperative DLCO mean $\pm$ SD (data not available for 24 patients) | $71.3 \pm 20.1$ |
| Histology: | |
|    Adenocarcinoma | 90 (32.1%) |
|    Squamous cell carcinoma | 120 (42.9%) |
|    Carcinoid | 30 (10.7%) |
|    Small cell carcinoma | 11 (3.9%) |
|    Other | 29 (10.4%) |
| Neoadjuvant chemotherapy | 119 (42.5) |

SD = standard deviation.

Preoperative chemotherapy was administered in 119 (42.5%) patients, and chemoradiotherapy in 7 cases (2.5%).

Surgery was performed on the right lung in 204 cases (72.8%). Right upper sleeve lobectomy was the main operation performed in 162 cases (57.9%), followed by left upper sleeve lobectomy (22.9%), left lower sleeve lobectomy (3.9%), and middle lobectomy (2.8%). Twenty-seven patients (9.6%) underwent right upper sleeve bilobectomy, and only two patients underwent right lower bilobectomy.

Vascular sleeves were performed in 41 cases (14.7%). One hundred and six patients (37.9%) underwent extended resections to extra parenchymal structures, most of which were extended to the pulmonary artery (64 cases; 22.9%) and vena cava (24 cases; 8.5%). The bronchial resection technique included circumferential resection in 250 cases (89.2%) and wedge resection in 20 cases (7.1). Bronchoplasty was performed with a continuous suture in 262 patients (93.6%) and with interrupted stitches in 9 patients (3.2%). For nine patients, this information was not available from the operatory reports. Overall morbidity in our series was 41.9%, and all complications were classified following the Clavien–Dindo

classification. Grade I complications occurred in 8.2% of cases, grade II in 20.7% of cases, grade III in 4.3% of cases, and more severe grade IV in 7.5% of cases. Mortality at 30 days was 3.9%. Table 2 describes all surgical results.

**Table 2.** Surgical results.

| Procedures | Patients *n* (%) |
|---|---|
| Side | |
| Right | 204 (72.8%) |
| Left | 76 (27.2%) |
| Intervention | |
| Right upper sleeve lobectomy | 162 (57.9%) |
| Left upper sleeve lobectomy | 64 (22.9%) |
| Right lower sleeve lobectomy | 0 |
| Left lower sleeve lobectomy | 11 (3.9%) |
| Right upper sleeve bilobectomy | 27 (9.6%) |
| Right lower bilobectomy | 2 (0.7%) |
| Medium lobectomy | 8 (2.8%) |
| Other | 6 (2.1%) |
| Type of sleeve resection | |
| Bronchial | 239 (85.4%) |
| Broncho-vascular (pulmonary artery) | 41 (14.7%) |
| Extended resection | |
| Pulmonary artery plastic | 64 (22.9%) |
| Vena cava resection | 24 (8.5%) |
| Azygous vein resection | 13 (4.6%) |
| Chest wall | 1 (0.4%) |
| Phrenic nerve | 3 (1.1%) |
| Pericardium | 1 (0.4%) |
| Bronchial resection technique (Data not available for 10 cases) | |
| Wedge resection | 20 (7.14%) |
| Circumferential resection | 250 (89.2%) |
| Bronchoplastic technique (data not available for 9 cases) | |
| Continuous suture | 262 (93.6%) |
| Interrupted suture | 9 (3.2%) |
| Morbidity (30 days) | 41.9% |
| Mortality (30 days) | 3.92% |

Following the eighth edition of the NSCLC TNM Classification, 60 patients were at stage I (21.4%) and 85 were at stage II (30.4%). Most of the patients were at stage III (42.9%). The staging was impossible for six patients because the tumors were secondary. At the final pathological exam, nine patients were without vital tumors due to previous treatments (3%).

One hundred and six patients were pN0 (37.9%) on the histologic specimen, 92 patients were pN1 (31.8%), and 82 patients were pN2 (29.3%). Complete radical resection was obtained in 263 cases (93.9%), while 17 (6.1%) resulted in R1 surgery.

*3.1. Anastomotic Complications and Outcome of Treatments*

Table 3 shows anastomotic complications that occurred in 18 patients (6.4%): late stenosis in 11 patients (3.9%), early and late broncho-pleural fistula (BPF) in 5 patients (1.8%), and early broncho-vascular fistula (BVF) in 2 patients (0.7%).

Late stenosis was treated with the endoscopic procedure, using a YAG laser for disostruction in rigid bronchoscopy. Only one case needed the implantation of a Dumon stent.

Early surgical treatment was the option chosen for four patients diagnosed with broncho-pleural fistula. Three patients underwent pneumonectomy, and two died of

complications such as sepsis and respiratory failure. One patient successfully underwent a re-thoracotomy for direct suture of the anastomosis.

**Table 3.** Anastomotic complications.

| Complication | Number (%) |
|---|---|
| Anastomotic complications | |
|     Yes | 18 (6.4%) |
|     No | 262 (93.6%) |
| Anastomotic stenosis | |
|     Early (<30 days) | 0 |
|     Late (>30 days) | 11 (3.9%) |
| Broncho-pleural fistula | |
|     Early (<30 days) | 4 (1.4%) |
|     Late (>30 days) | 1 (0.4%) |
| Arterio-bronchial fistula | |
|     Early (<30 days) | 2 (0.7%) |
|     Late (>30 days) | 0 |
| Surgery | 4 (1.4%) |
|     Interventional bronchoscopy | |
|     Stenting | 1 (1.4%) |
|     Dilatation (laser/mechanical) | 11 (3.9%) |
|     Glue | 1 (1.4%) |

Only one patient with a bronchial fistula was effectively treated with glue installation using a flexible bronchoscope.

Patients affected by arterio-bronchial fistula rapidly died of massive hemoptysis and acute hemodynamic failure without the chance to treat this lethal condition.

Management of anastomotic complications for all patients is summarized in Table 4.

**Table 4.** Management of anastomotic complications.

| Patient no. | Sex | Age | Surgery | Type of Sleeve | Stage | Anastomosis Complication | Treatment | OS (Months) |
|---|---|---|---|---|---|---|---|---|
| 1 | M | 67 | RB | BS | IIIB | Late stenosis | RBALA | 163.4 |
| 2 | M | 71 | LLL | BS | IIIB | Late stenosis | RBALA | 78.4 |
| 3 | M | 52 | RUL | BS | IIIA | Early broncho-pleural fistula | Pneumonectomy | 0.7 |
| 4 | M | 58 | LUL + PA | ABS | IIIA | Early broncho-pleural fistula | Endo-bronchial glue | 9.6 |
| 5 | M | 56 | RUL | BS | IIIA | Early broncho-pleural fistula | continuous suture | 76.7 |
| 6 | F | 50 | RUL | BS | IA | Late stenosis | RBALA | 105.8 |
| 7 | M | 63 | RUL | BS | IIA | Late stenosis | RBALA | 29.1 |
| 8 | F | 46 | RUL | BS | IIIB | Early broncho-pleural fistula | Pneumonectomy | 36.7 |
| 9 | F | 73 | RUL | BS | IA | Early arterio-bronchial fistula | Surveillance | 1.2 |
| 10 | M | 65 | RUL | BS | IIB | Late stenosis | RBALA | 65.9 |
| 11 | F | 56 | LUL + PA | ABS | IIA | Late stenosis | RBALA | 78.9 |
| 12 | F | 61 | RUL | BS | IIA | Late stenosis | RBALA | 25.7 |
| 13 | M | 68 | RUL | BS | IIIA | Late stenosis | RBALA | 13.2 |
| 14 | M | 20 | RB | BS | IIIA | Late stenosis | RBALA | 69.3 |
| 15 | M | 76 | LUL | BS | IIA | Late stenosis | RBALA | 57.4 |
| 16 | M | 66 | RUL + VC | BS | IIIB | Late stenosis | RBALA + stenting | 21.3 |
| 17 | F | 75 | RUL | BS | IB | Early broncho-pleural fistula | Pneumonectomy | 0.9 |
| 18 | M | 55 | RUL + AV | BS | IIIB | Early arterio-bronchial fistula | Not appliable | 0.6 |

RB = right bilobectomy; LLL = left lower lobectomy; RUL = right upper lobectomy; LUL = left upper lobectomy; PA = pulmonary artery; VC = vena cava; AV = azigos vein; BS = bronchial sleeve; ABS = arterio-bronchial sleeve; RBALA = rigid bronchoscopy associated to laser ablation.

### 3.2. Outcome and Overall Survival

Figures 1 and 2 represent survival curves.

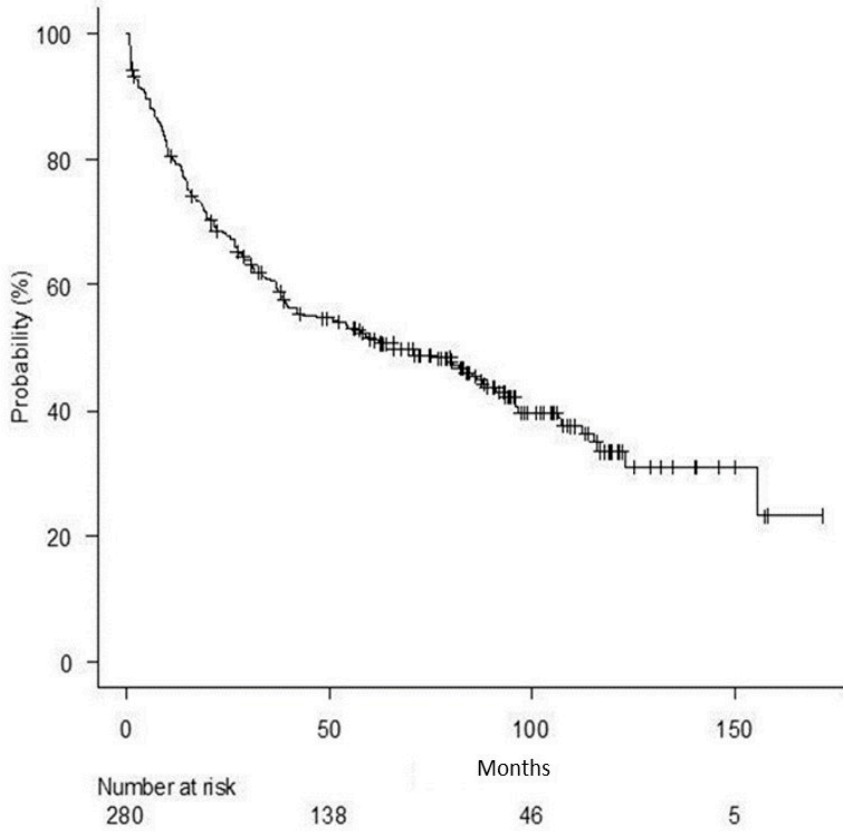

**Figure 1.** Overall survival for all patients.

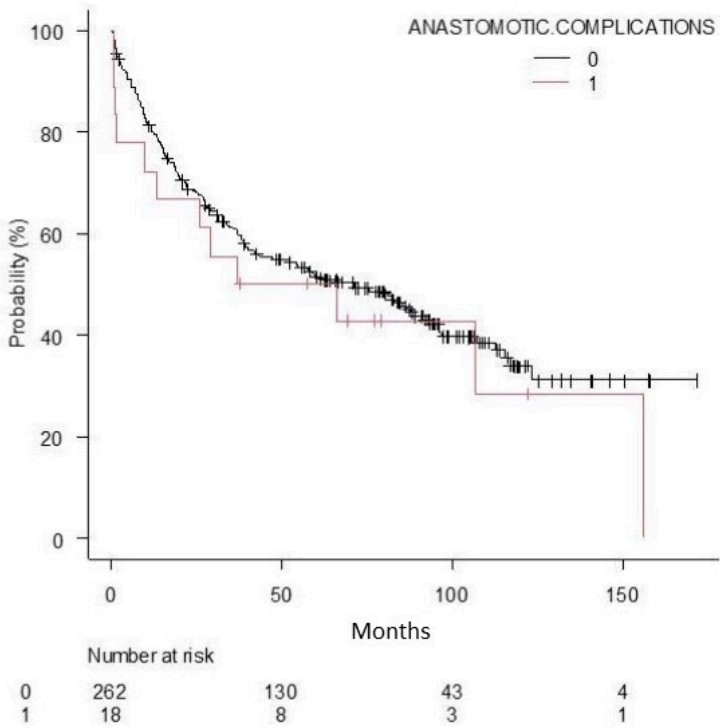

**Figure 2.** Compared overall survivals based on anastomotic complication.

All patients' median overall survival (OS) was 65.90 months (95% CI = 41.8–91.0).

There were no statistically significant differences ($p = 0.38$) in the survival of patients without anastomotic complication (71.0 months) and those with anastomotic complication (51.3 months).

We performed a univariable analysis to establish factors conditioning anastomotic complication, which we summarized in Table 5. We observed a statistically significant impact of cardiovascular history ($p = 0.044$), neoadjuvant chemo-radiotherapy ($p = 0.023$), nodal stage ($p = 0.0023$), and radical dissection ($p = 0.025$).

**Table 5.** Univariable analysis.

| Characteristic | Complicated (no. = 18) | Not Complicated (no. = 262) | *p*-Value |
|---|---|---|---|
| Age mean ± SD | 60 ± 13.2 | 62.4 ± 11.4 | 0.65 |
| Smoking history | 12 (66.7) | 218 (83.2) | 0.068 |
| Cardiovascular history | 8 (44.4) | 90 (34.45) | 0.044 |
| Pulmonary history | 4 (22.2) | 56 (21.37) | 0.82 |
| Neoadjuvant chemotherapy | 10 (55.6) | 109 (41.6) | 0.054 |
| Neoadjuvant chemo-radiotherapy | 1 (5.6) | 6 (2.3) | 0.023 |
| Pathological staging | | | |
| N0 | 3 (16.6) | 103 (39.3) | |
| N1 | 11 (61.1) | 81 (30.9) | 0.023 |
| N2 | 4 (22.2) | 78 (29.8) | |
| Radical dissection | | | |
| R0 | 15 (83.3) | 248 (94.7) | 0.025 |
| R1 | 3 (16.7) | 14 (5.3) | |
| Side | | | |
| Right | 14 (77.8) | 190 (72.5) | 0.51 |
| Left | 4 (22.2) | 72 (27.5) | |
| Intervention | | | |
| Right upper sleeve lobectomy | 12 (66.7) | 150 (57.3) | |
| Left upper sleeve lobectomy | 3 (11.1) | 61 (23.3) | |
| Right lower sleeve lobectomy | 0 | 0 | |
| Left lower sleeve lobectomy | 1 (5.6) | 10 (3.8) | 0.43 |
| Right upper sleeve bilobectomy | 2 (11.1) | 25 (9.5) | |
| Right lower bilobectomy | 0 | 8 (3.1%) | |
| Medium lobectomy | 0 | 6 (2.3%) | |
| Other | | | |
| Type of sleeve resection | | | |
| Bronchial | 16 (88.9) | 223 (85.2) | 0.45 |
| Broncho-vascular (pulmonary artery) | 2 (11.1) | 39 (14.8) | |
| Extended resection | | | |
| Pulmonary artery plastic | 2 (11.1) | 62 (23.6) | |
| Vena cava resection | 1 | 23 (8.7) | |
| Azygous vein resection | 1 (5.6) | 12 (4.5) | 0.19 |
| Chest wall | 0 | 3 (1.1) | |
| Phrenic nerve | 0 | 3 (1.1) | |
| Pericardium | 0 | 1 (0.38) | |
| Bronchial resection technique (data not available in 10) | | | |
| Wedge resection | 1 (11.1) | 19 (7.2) | |
| Circumferential resection | 16 (88.9) | 234 (89.3) | 0.33 |
| NA | 1 | 9 (3.4) | |
| Mortality (30 days) | 16.7% | 3.0% | 0.013 |

Mortality at 30 days was dramatically higher in patients with anastomotic complications (16.7% vs. 3%, *p* = 0.014). Hospital mortality observed for these patients was 11%.

The multivariable analysis confirmed that only the pathological stage (N+) is a risk factor for anastomotic complications (*p* = 0.00164), as shown in Table 6.

**Table 6.** Multivariable analysis.

| | OR | 95% CI | *p*-Value |
|---|---|---|---|
| Age mean ± SD | 0.97 | 0.93–1.03 | 0.39 |
| Smoke | 0.40 | 0.08–1.03 | 0.18 |
| Cardiovascular history | 1.43 | 0.42–4.84 | 0.17 |
| Pulmonary history | 1.26 | 0.33–3.93 | 0.94 |
| Neoadjuvant chemotherapy | 1.90 | 0.62–6.14 | 0.18 |
| Neoadjuvant chemo-radiotherapy | 1.87 | 0.54–5.17 | 0.31 |
| Pathological staging (N1) | 6.02 | 1.67–9.31 | 0.016 |
| Radical dissection | 5.72 | 0.96–9.44 | 0.091 |
| Side | 0.56 | 0.23–3.42 | 0.96 |
| Intervention | 0.48 | 0.34–1.55 | 0.88 |
| Type of sleeve resection | 0.54 | 0.34–2.56 | 0.68 |
| Extended resection | 0.32 | 0.054–1.36 | 0.77 |
| Bronchoplasty resection technique | 0.78 | 0.55–2.48 | 0.90 |

## 4. Discussion

Bronchial sleeve resection is considered a safe surgical option for lung-sparing resection regardless of respiratory function.

This technique carries a postoperative mortality rate of 1.5% to 11.0% and an overall morbidity rate of 11.0% to 51.0% [1,5–7,9,13–17,20–23].

Our study presents a mortality rate of 3.4% according to previous experiences, and a morbidity rate of 41.8% considering all complications.

Surgery-specific complications related to airway anastomosis include the development of benign anastomotic strictures, broncho-pleural or broncho-vascular fistulas, and local tumor recurrence.

According to recent series results, anastomotic complications were seen in 6.4% of our patients. Mortality at 30 days was significantly higher in those with anastomotic complications (*p* = 0.013).

Yildizeli and colleagues [15] found a rate of 6.4% of anastomotic complications, with numbers close to the series reported (14 cases in 218 patients operated on).

In the Blicki et al. [24] study, the complication rate was 21.3% (23 of 108 patients), which is exceptionally high when compared with previous studies. Authors correlated this result with the broad experience in interventional endoscopy, which would determine an overdiagnosis of bronchial complications.

BPFs are described in the literature in 0% to 9.3% of patients who underwent sleeve resections [23]. In our study, it occurred in five cases (1.8% of all patients). Four patients who underwent right upper lobectomy showed early BPF, and one patient who underwent left upper lobectomy developed the complication late.

Right pneumonectomy was performed in two cases: one patient was successfully treated endoscopically with glue injection and the other underwent right tracheal sleeve pneumonectomy after unsuccessful Dumon stent placement. The patient with the left fistula underwent re-operation with a direct suture of the defect.

In our experience, surgery, if possible, is preferred to avoid life-threatening complications due to empyema and sepsis. Nevertheless, two patients who underwent pneumonectomy died of complications.

New frontiers of regenerative medicine are under investigation, and we expect they will offer alternative treatments [32].

BVFs occur in 0% to 2.5% of patients after sleeve resections [23]. They are rare but almost lethal. These patients typically manifest with massive hemoptysis and acute respiratory and hemodynamic failure.

A higher risk of BVF has been observed in patients undergoing combined airway and vascular resections and reconstructions, and mortality is close to 100%.

The prevention of fistulas is a goal of the surgical technique that should aim to preserve bronchial vascularization and reduce the tension on the anastomosis. We also consider the coverage of the suture with a vital tissue flap to separate the anastomosis from vascular structures and avoid decubitus.

In the hypothesis of this complication, which frequently occurs in the first 30 days, a flexible inspective bronchoscopy should be performed in all patients before discharge.

Benign lumen stenosis results from ischemia at the anastomosis leading to granuloma formation and has been described in 0% to 15.1% of patients undergoing sleeve resections [23]. It was observed in 11 cases (3.92%) in our series.

Bronchoscopic balloon dilation is the treatment of choice in patients with benign anastomotic strictures, and endoscopic procedures can be repeated if necessary [24].

In particular cases, we should consider endobronchial stents, even if their applications are more difficult after sleeve lobectomy.

In our report, pathological lymph nodal staging is significantly related to the risk of complications ($p = 0.014$), as confirmed by multivariable analysis. In our opinion, it reflects the aggressiveness of surgical maneuvers in eradicating cancer which requires extended radical lymphadenectomy with damage to bronchial vascularization.

The tissue damage and the compromised vascularization may cause the potential healing impairment of the reconstructed bronchus.

In the literature, nodal involvement is considered a determinant of short- and long-term survival after sleeve resection. Schirren et al. [33] studied the role of sleeve resection in advanced nodal disease.

Lymph node involvement is an adverse prognostic factor in survival, but sleeve resections do not result in higher morbidity and mortality.

Beyond the results of the individual series, we cannot ignore the weight of lymph node and mediastinal involvement, which can be decisive in the surgical outcome.

Rendina et al. [20] suggested that risks for complications may be related to the increased difficulty in surgical dissection caused by the fibrotic reaction that follows the induction therapy that is typically administered.

In our multivariable analysis, neoadjuvant treatments were not a risk factor for anastomotic complications (chemotherapy $p = 0.18$; chemo-radiotherapy $p = 0.31$).

Different studies evaluated the impact of neoadjuvant treatments on sleeve resection outcomes in the literature.

Veronesi et al. reported a low morbidity rate, no mortality, and no anastomotic dehiscence associated with preoperative treatment, confirming that sleeve resection after induction treatment is feasible and safe [23].

In the experience of Rea and associates [16] regarding sleeve lobectomy, the univariate analysis only identified neoadjuvant radiation therapy as predictive for early bronchial complications.

Moreover, Gonzales et al. [18] observed no impact of induction therapy on airway complications after sleeve lobectomy.

Bagan et al. [27] reported that complication rates in the induction chemotherapy and surgery-only groups were 23.8% and 24.7%, respectively. Major anastomotic complications—

one broncho-pleural fistula and two severe stenoses—were observed in three patients treated only with surgery.

In contrast, Milman and colleagues [14] reported a study where airway complications occurred only in patients who underwent induction therapy, with an incidence of 4.7%.

## 5. Conclusions

Bronchoplasty is a safe procedure that allows for parenchymal-sparing resection while maintaining low mortality and morbidity.

Despite this, mortality rate at 30 days increases significantly in the case of anastomotic complications.

Risk factors like cardiovascular disease and advanced nodal staging have been identified.

For this reason, during patient selection, we must carefully consider comorbidities and previous treatments to better plan surgery.

In our experience, surgery is preferred to avoid life-threatening complications in broncho-pleural fistulas. Bronchoscopic balloon dilatation is the treatment of choice for benign strictures.

The early diagnosis of anastomotic complication is important, and it is possible with bronchoscopy performed routinely.

Finally, for their management, an expert team trained in advanced surgical methods and interventional bronchoscopy techniques is essential.

**Author Contributions:** L.G. (Conceptualization; Data curation; Investigation; Writing—original draft; Writing—review and editing); L.B. (Formal analysis; Supervision; Writing—original draft); M.C. (Investigation); F.P. (Investigation); D.G. (Investigation); A.M. (Investigation); S.D. (Data curation; Methodology); G.L.I. (Data curation); A.C. (Data curation); J.G. (Methodology; Supervision); L.S. (Supervision; Visualization; Writing—review and editing). All authors have read and agreed to the published version of the manuscript.

**Funding:** This work was partially supported by the Italian Ministry of Health with Ricerca Corrente and 5 × 100 funds.

**Institutional Review Board Statement:** The study was conducted according to the Declaration of Helsinki guidelines. Ethical review and approval were not required for retrospective study on human participants in accordance with the local legislation and institutional requirements. The activities carried out are part of regular care activities and the patients are sufficiently informed through institutional information and consent forms.

**Informed Consent Statement:** The study was conducted according to the guidelines of the Declaration of Helsinki; the Ethics Committee of our Institution waived the need for ethics approval and the need to obtain consent for the collection, analysis, and publication of the retrospectively obtained and anonymized data for this non-interventional study. Each patient in the study signed standard consents to clinical treatment and to processing of personal data for scientific research.

**Data Availability Statement:** The data presented in this study are available on request from the corresponding author. The data are not publicly available due to privacy restriction local legislation.

**Acknowledgments:** The authors wish to acknowledge Daniela Brambilla and Giulia Sedda, Division of Thoracic Surgery, IEO, European Institute of Oncology IRCCS, Milan, Italy.

**Conflicts of Interest:** The authors declare no conflict of interest.

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
