# Peer review of "Anastomosis Complications after Bronchoplasty: Incidence, Risk Factors, and Treatment Options Reported by a Referral Cancer Center"

_curroncol, doi:10.3390/curroncol30120760_

Round 1

Reviewer 1 Report

Comments and Suggestions for Authors

Congratulations to you on your successful operation of sleeve resections on lung cancers. The entire manuscript was well written except it needs to be revised and further data provided in some places. The following are my comments.

(1) Your population included small cell carcinoma. Was there any difference between the small cell and non-small cell cancer on survival and complications ?

(2) I can't see the data of local recurrence.

(3) Moreover, the is no data of bronchomalacia.

(4) Twenty six out of 35 references are over ten years. I suggest you should re-new.

Author Response

(1) Your population included small cell carcinoma. Was there any difference between the small cell and non-small cell cancer on survival and complications?

-As reported in our paper “Small cell carcinoma resulted in 11 patients (3.9%). These patients were operated on from 1998 till 2008 with a different preoperative diagnosis.”

Due to the low incidence of this histological subtype we didn’t consider of interest for our analysis.

Moreover, "Histology" is a variable reported with a descriptive intent on population but the focus of our study is on surgical and technical aspects.

Finally, none patients with small cell carcinoma had anastomotic complications after surgery.

(2) I can't see the data of local recurrence.

- I’m sorry, but we didn’t collect this data.

(3) Moreover, the is no data of bronchomalacia.

-There’s no data on bronchomalacia because this event was not reported in our experience.

(4) Twenty six out of 35 references are over ten years. I suggest you should re-new.

-Indeed, incorporating recent reference updates is critical to enhance the scholarly integrity and relevance of the work. We have integrated the latest and most pertinent references to ensure that the content remains aligned with the current state of knowledge in the field as you already suggested. However, there's no more recent literature on the main focus of our article, that is the incidence and the management of complications of bronchial anastomosis after lung surgery

Reviewer 2 Report

Comments and Suggestions for Authors

Dear Authors and Editor

 I had the pleasure to review this interesting article entitled “Anastomosis Complications after Bronchoplasty: incidence, risk factors and treatment options reported by a referral Cancer Center”.

Herein, authors described their astonishing experience in management of bronchoplasty complications.

The manuscript is very interesting and well-written with just some minor issues and concerns that could be issued.

1.       Minor typos and english grammar errors should be corrected.

2.       Introduction includes different paragraphs belonging to the material and methods and discussion.

3.       In M&M preoperative assessment could be better described. In some reported cases, pre-operative EBUS is not routinely recommended.

4.        Cases of pR1 were intraoperatively evaluated with a frozen section?

5.       Have the authors experienced cases of unplanned bronchoplasty (required for unexpected intraoperative findings?). have the authors compared these two groups?

6.       Besides the transposition of a pedicel flap, have the authors ever used other sealant devices ?

Comments on the Quality of English Language

   Minor typos and english grammar errors should be corrected.

Author Response

1.Minor typos and english grammar errors should be corrected.

-I’ve checked and corrected grammar errors.

2. Introduction includes different paragraphs belonging to the material and methods and discussion.

I’m sorry but I don’t understand your observation. Maybe some concepts are reiterated in the different sections. If you have noticed the repetition of some paragraphs that has escaped our control, please report them so I can make the appropriate modifications.

3. In M&M preoperative assessment could be better described. In some reported cases, pre-operative EBUS is not routinely recommended.

In the last 30 years, imaging methods have been very advanced, as biopsy techniques are supported by technologies that make procedures much less invasive. In the past, histological staging of the mediastinum was performed in case of radiological evidence of N2 station involvement using surgical procedure of mediastinoscopy. Recently the Endobronchial Ultrasound (EBUS), has been employed in our Unit for the tumor with central envolvement. This minimally invasive procedure is useful both for diagnosis, mediastinal staging but also for the study of the anatomy or local infiltration. For this reason in pre operative planning ,when a bronchoplasty is expected, we always perform an evaluation with EBUS.

4. Cases of pR1 were intraoperatively evaluated with a frozen section?

In our study 17 patients (6.1%) resulted in R1-surgery. We perform frozen section in case of suspected infilitration of resection margins, evidenced during the surgery. In case of infiltration we proceed to obtain radicality. These 17 cases reported, resulted R1 only at the specimen examination, because of microscopic invasion and more frequently for a growth of the cancer in the sub mucosal layer

5. Have the authors experienced cases of unplanned bronchoplasty (required for unexpected intraoperative findings?). have the authors compared these two groups?

Surgery was well planned with pre-operative imaging and bronchoscopy. Unexpected bronchoplasty was not reported in our experience. 

6. Besides the transposition of a pedicel flap, have the authors ever used other sealant devices ?

-In the recent years we introduced in our surgery sealant devices to reinforce hemostasis and air leaks control but not in replace of pedicle flap.

Reviewer 3 Report

Comments and Suggestions for Authors

Thank you for the opportunity to analyze your interesting article.             

In this article, authors have analyzed bronchoplastic procedures short-term and long-term results with a focus on anastomotic complications in their unit. 

            Concerning the introduction:

            The introduction is well written, introducing indications of bronchoplastic resection, advantages compared to pneumonectomy, new challenges of this kind of surgery after systemic therapies and radiotherapies, the possibility to preform this surgery with a minimally invasive approach and the challenging situation which are anastomotic complications. No major concerns. 

            Concerning the methodology:

            Population:

            No major concerns, but since 1998 pre operative management of patients has evolved, especially pre operative stagging. 

            Maybe the upstaging can be analyzed by year and compared to the clinical stagging, EBUS, mediastinoscopy, or only CT scan and PET scan. 

            How many patients were pre operatively re stagged by mediastinoscopy after neo adjuvant therapy? 

            Surgery:  

            Maybe some surgical pictures can be interesting         

Statistical analysis conducted: 

            No major concerns about it. 

            Concerning the results

            Results are well reported and clearly presented in the article and in the table.

-       Concerning the surgical procedures, can you also report

o    the kind of bronchial coverage performed at the end of the procedure? Muscle / pericardial / Fat flap ?  

o   Surgical duration

o   Peroperative blood loss

-       Concerning comorbidities, do you have more details about diabetes? Stroke? Coronaropathy? More details concerning cardiovascular history? Previous treatment with corticosteroids? Immune diseases? High blood pressure is different from coronaropathy for example - it’s described as anastomotic complications risk factors. 

-       Concerning the pathological results:

o   Do you have the T and N up stagging? 

o   A table can be more informative than the text. 

-       Concerning the mortality, 90-day is more informative than 30-day mortality today.

-       Concerning the anastomotic complications

o   How was diagnosed patients with an early of late BPF? There were clinical symptoms or it was during the scheduled bronchoscopy? 

o   Same question for late anastomotic stenosis. 

o   The table 4 can be re organized according complications: BPF, early, late, arterio-bronchial fistula, anastomotic stenosis. This will be easiest to read. 

o   What kind of coverage was performed during re-intervention?
What was the glue used during the endoscopic procedure? Because on mucosal layers, “nothing works” “usually”.

o   This is not described but during your endoscopic protocol follow-up, how many patients have presented bronchial ischemia lesions without fistula? 

-       Concerning the risk factors, because you have good short-term and long-term results, your risks factors are in accordance with the literature. Is there a trend with less complications in the 5 last years or the rate of fistula and stenosis is stable over the period? 

            Concerning the discussion:

It’s a well written discussion well documented with good references. 

The literature are well presented, with epidemiology and main risks factor.

Limitations are well described.

Concerning the conclusion:

            It’s a well written, easy reading and interesting article, that need some precisions nevertheless.

Finally, few complications occurred and you may have some tips and tricks to prevent this complication.

Maybe you should bring more details about your patients’ selection today, your prevention with a flap to cover the sleeve, will be very interesting.

            Except these remarks, congratulations to authors for this work. 

Author Response

In this article, authors have analyzed bronchoplastic procedures short-term and long-term results with a focus on anastomotic complications in their unit.

Concerning the introduction:

The introduction is well written, introducing indications of bronchoplastic resection, advantages compared to pneumonectomy, new challenges of this kind of surgery after systemic therapies and radiotherapies, the possibility to preform this surgery with a minimally invasive approach and the challenging situation which are anastomotic complications. No major concerns.

Concerning the methodology:

Population:

No major concerns, but since 1998 pre operative management of patients has evolved, especially pre operative stagging.

Maybe the upstaging can be analyzed by year and compared to the clinical stagging, EBUS, mediastinoscopy, or only CT scan and PET scan.

-I’m agree that in the last 30 years, imaging methods have been advanced, as biopsy techniques are supported by technologies that make procedures much less invasive. In the past, histological staging of the mediastinum was performed in case of radiological evidence of N2 station involvement using surgical procedure of mediastinoscopy. Recently the Endobronchial Ultrasound (EBUS), has been employed in our Unit for the tumor with central envolvement. This minimally invasive procedure is useful both for diagnosis, mediastinal staging but also for the study of the anatomy or local infiltration.

However the upstaging changing and the comparison between clinical and surgical staging was not in our endopoints and data were not analyzed because uncomplete.

How many patients were pre operatively re stagged by mediastinoscopy after neo adjuvant therapy?

A pre-operative invasive re-staging was not routinely performed, and this data was not collected.

Surgery:

Maybe some surgical pictures can be interesting.

I can find and attach if the Journal needs.

Statistical analysis conducted:

No major concerns about it.

Concerning the results

Results are well reported and clearly presented in the article and in the table.

- Concerning the surgical procedures, can you also report

o the kind of bronchial coverage performed at the end of the procedure? Muscle / pericardial / Fat flap

Different surgeons used pedicle flap of pericardial fat, pericardium, or pleura dependig on anatomy and patients conditions. This data was not collected from the operatory report.

o Surgical duration. I'm sorry but this data was not collected

o Peroperative blood loss.  I'm sorry but this data was not collected

- Concerning comorbidities, do you have more details about diabetes? Stroke? Coronaropathy? More details concerning cardiovascular history? Previous treatment with corticosteroids? Immune diseases? High blood pressure is different from coronaropathy for example - it’s described as anastomotic complications risk factors.

We resumed in “cardiovascular disease” pathologies as coronaropathy, hypertension, heart disease, Other pathologies were not considered in our data collections

Concerning the pathological results:

· Do you have the T and N up stagging?

The 8th edition of NSCLC TNM Classification was applied on final pathological staging. The upstaging from clinical pre-operative stage was not evaluated because was not included in our analysis. 

- Concerning the mortality, 90-day is more informative than 30-day mortality today.

We chose to consider 30-day mortality because was more relevant for patients with anastomotic complications

- Concerning the anastomotic complications

o How was diagnosed patients with an early of late BPF? There were clinical symptoms or it was during the scheduled bronchoscopy?

All patients with broncho-pleural fistula manifested common clinical symptoms like cough, fever and subcutaneous emphysema. Early diagnosis of necrosis is possible during scheduled bronchoscopy and can suggest that the anastomosis is suffering and BPF can occur but not necessary evolves .

Patients with

· Same question for late anastomotic stenosis.

Stenosis was diagnosed during the scheduled bronchoscopy, not for clinical manifestations

· The table 4 can be re organized according complications: BPF, early, late, arterio-bronchial fistula, anastomotic stenosis. This will be easiest to read.

The table 4 is organized according with the date of complication and its management. If strongly suggested the order can be changed.

o What kind of coverage was performed during re-intervention? What was the glue used during the endoscopic procedure? Because on mucosal layers, “nothing works” “usually”.

· This is not described but during your endoscopic protocol follow-up, how many patients have presented bronchial ischemia lesions without fistula?

I’m sorry but we don’t have this data.

- Concerning the risk factors, because you have good short-term and long-term results, your risks factors are in accordance with the literature. Is there a trend with less complications in the 5 last years or the rate of fistula and stenosis is stable over the period?

In the last 5 years we observed a lower rate of anastomotical complications in sleeve resection, but we could open a large discussion on the rate of fistula in lung surgery. This  is another important topic related to new techniques, advances in treatments and age of patients

Concerning the discussion:

It’s a well written discussion well documented with good references.

The literature are well presented, with epidemiology and main risks factor.

Limitations are well described.

Concerning the conclusion:

It’s a well written, easy reading and interesting article, that need some precisions nevertheless.

Finally, few complications occurred and you may have some tips and tricks to prevent this complication.

Maybe you should bring more details about your patients’ selection today, your prevention with a flap to cover the sleeve, will be very interesting.

Except these remarks, congratulations to authors for this work.